# Relationship between Antenatal Mental Health and Facial Emotion Recognition Bias for Children’s Faces among Pregnant Women

**DOI:** 10.3390/jpm12091391

**Published:** 2022-08-27

**Authors:** Youji Takubo, Naohisa Tsujino, Yuri Aikawa, Kazuyo Fukiya, Takashi Uchino, Naoyuki Katagiri, Megumu Ito, Yasuo Akiba, Masafumi Mizuno, Takahiro Nemoto

**Affiliations:** 1Department of Neuropsychiatry, Toho University Graduate School of Medicine, 5-21-16 Omori-nishi, Ota-ku, Tokyo 143-8540, Japan; 2Department of Psychiatry, Saiseikai Yokohamashi Tobu Hospital, 3-6-1 Shimosueyoshi, Tsurumi-ku, Yokohama 230-8765, Japan; 3Department of Neuropsychiatry, Toho University Faculty of Medicine, 6-11-1 Omori-nishi, Ota-ku, Tokyo 143-8541, Japan; 4Department of Obstetrics and Gynaecology, Saiseikai Yokohamashi Tobu Hospital, 3-6-1 Shimosueyoshi, Tsurumi-ku, Yokohama 230-8765, Japan; 5Tokyo Metropolitan Matsuzawa Hospital, 2-1-1 Kamikitazawa, Setagaya-ku, Tokyo 156-0057, Japan

**Keywords:** antenatal, bonding failure, depression, facial emotion recognition, perinatal, social cognition

## Abstract

The importance of identification of facial emotion recognition (FER) bias for a child’s face has been reinforced from the perspective of risk screening for future peripartum mental health problems. We attempted to clarify the relationship of FER bias for children’s faces with antenatal depression and bonding failure among pregnant women, taking into consideration their broad social cognitive abilities and experience in child raising. This study had a cross-sectional design, and participants were women in their second trimester of pregnancy. Seventy-two participants were assessed by the Edinburgh Postnatal Depression Scale (EPDS), the Mother-to-Infant Bonding Questionnaire (MIBQ), and a series of social cognitive tests. FER bias for a child’s face was assessed by Baby Cue Cards (BCC), and a larger number of disengagement responses suggest greater sensitivity to a child’s disengagement facial expressions. In a regression analysis conducted using EPDS as the dependent variable, a larger number of disengagement responses to the BCC (β = 0.365, *p* = 0.001) and the primipara status (β = −0.263, *p* = 0.016) were found to significantly contribute to antenatal depressive symptoms. Also, more disengagement responses to the BCC also significantly contributed to bonding failure as measured by the MIBQ (β = 0.234, *p* = 0.048). Maternal sensitivity to the child’s disengagement cues was associated with antenatal depressive symptoms and bonding failure more than the other social cognitive variables. The effects of FER bias on postpartum mental health and abusive behavior needs to be clarified by further longitudinal studies.

## 1. Introduction

Depression is the most prevalent morbidity during the peripartum period [1]. According to a meta-analysis of Japanese studies, the prevalence of depression in the second trimester, third trimester, and first month postpartum were 14.0%, 16.3%, and 15.1%, respectively [2]. Numerous studies have demonstrated risk factors for peripartum depression including a history of affective disorders, psychosocial factors, and bonding failure [3,4]. Bonding failure is characterized by an aversion to the child and marked impairment in interactions with the child [5], and the coexistence of bonding failure and peripartum depression has also been attracting attention [6,7]. Because a child’s cues are largely non-verbal, the mother’s ability to process the child’s facial expressions and respond appropriately to social cues are crucial for mother-child interactions and maternal mental well-being [8]. Meanwhile, a study revealed that changes in the cerebral neural network in areas subserving social cognition occur during pregnancy and childbirth, suggesting an adaptive process to motherhood [9]. In these contexts, much research in recent years has focused on social cognition, particularly facial emotion recognition (FER) of peripartum mothers.

A meta-analysis suggested that deficits in FER are associated with major depressive disorder in the general population [10]. Besides, this review revealed that less accurate recognition of happy facial expressions is associated with higher levels of severity of depressive symptoms [10]. With regard to peripartum women, mothers with postpartum depression tend to be less sensitive to their child’s social cues when interacting with children [11]. Moreover, there was a difference in the attentional bias to a child’s emotions between depressed and non-depressed pregnant women [12]. Because facial expression of emotions yet remain ambiguous and undifferentiated in childhood, the social cognitive process towards a child’s face is reported to be distinct from that towards an adult’s face [13]. In fact, pregnant women recognize happy faces and adults’ faces more quickly and accurately than sad faces and children’s faces, respectively [14]. However, most of the studies that have investigated the relationship between perinatal mental health and FER so far, have focused on only either FER for children’s faces or adults’ faces [15]. For example, mothers with postpartum depression were less likely to identify happy faces of children than non-depressed mothers [16]. Similarly, mothers with increased negative perception of a child’s emotions showed higher levels of anxiety in the early postpartum period [17]. Only a few studies have measured FER for both children’s and adults’ faces and their associations with maternal mental health [17,18]. A recent longitudinal study indicated that negative ratings for a child’s cries predicted enhanced risk of postpartum depression [18]. In addition, less efficient detection of a positive facial expression was associated with an increased risk for child abuse [19]. Therefore, the importance of identification of FER bias has been reinforced from the perspective of risk screening for future peripartum depression, bonding failure, and abusive behavior.

As for the Theory of Mind (ToM), that is, higher-order abilities for reasoning about others’ mental states, a few studies have explored the association between ToM and depressive symptoms in perinatal and non-perinatal samples [20,21]. However, the above-mentioned studies that investigated FER performance in perinatal mothers did not measure ToM [12,18]. Although parenting experience affects the FER performance for a child’s cues, very few studies in this field have focused on experience in child-raising [17]. Therefore, we took into consideration pregnant women’s parity (i.e., primipara or multipara) status for the analyses.

There is little evidence in regard to the relationships among antenatal depression, bonding failure, FER bias for children’s facial expressions, FER performance for adults’ facial expressions, ToM, and general social cognitive functions. The results of evaluation of these relationships in a cohort of pregnant women in their second trimester are expected to bridge the gap in the literature. The hypothesis of our study is that FER bias for children’s faces is more strongly associated with antenatal depressive symptoms and bonding failure than with other variables. In the present study, we attempted to clarify the relationship of FER bias for children’s faces with antenatal depression and bonding failure among pregnant women, taking broad social cognitive abilities and experience in child raising into consideration.

## 2. Materials and Methods

### 2.1. Procedures and Subjects

This study had a cross-sectional design, and the subjects were women in their second trimester (15–27 weeks) of pregnancy. The study period was from October 2020 to December 2021. The inclusion criteria for the subjects were: (a) established pregnancy; (b) mother tongue Japanese; and (c) no history of neurologic disorders, such as stroke, head trauma, or spinal cord injury. Subjects who were below the age of 20 years or had serious co-morbidity (e.g., fetal growth restriction, severe preeclampsia) were excluded. We recruited participants who had a series of prenatal check-ups at the Department of Obstetrics and Gynaecology of the Saiseikai Yokohamashi Tobu Hospital, which is a core hospital in the eastern part of Yokohama City. We obtained written informed consent from each of the subjects for participation in the study. The assessment was conducted in a quiet examination room to ensure the confidentiality of the participants. Participants were asked to complete a questionnaire pertaining to their background information and self-administered measures. Demographic and obstetric information, including the age, educational level, weeks of pregnancy, and parity status were collected from this questionnaire and medical charts. Some social cognitive tests were presented on an iPad Air 3 (Apple) and took, on average, 50 min. At the time of the social cognitive tests, the researchers conducted the tests without knowing the results of the self-administered measures. Because the physical condition of pregnant women in the second trimester is generally more stable than that in the first or third trimester, the results of assessment of their mental condition and cognitive skills may be less influenced by the physical condition in the second trimester.

This study was performed as part of the Mental health and Early Intervention in the Community-based Integrated care System (MEICIS) project, which is supported by a Health Labor Sciences Research Grant (19GC1015) to T.N. and N.T. In addition, this research was partly funded by the Inokashira Hospital Grants for Psychiatry Research in 2019 to Y.T. The procedures for this study were approved by the Ethics Committees of the Faculty of Medicine, Toho University (A20032), and the Saiseikai Yokohamashi Tobu Hospital (20200088). The study was performed in accordance with the principles laid down in the latest version of the Declaration of Helsinki (October 2013).

### 2.2. Measures

#### 2.2.1. Edinburgh Postnatal Depression Scale (EPDS) and Mother-to-Infant Bonding Questionnaire (MIBQ)

The states of antenatal depression and bonding failure were evaluated by the Edinburgh Postnatal Depression Scale (EPDS) and Mother-to-Infant Bonding Questionnaire (MIBQ) [22,23]. The EPDS and MIBQ consist of 10 items and 9 items, with the total scores ranging from 0 to 30, and 0 to 27, respectively. The Japanese versions of these two instruments have been validated for use among both pregnant women and postpartum mothers [24,25,26].

#### 2.2.2. Baby Cue Cards (BCC)

The FER bias for children’s facial expressions was assessed using Baby Cue Cards (BCC) [27]. The BCC comprises of 52 colored cards and each card represented a child’s social cue (e.g., hungry, affectionate, discomforting, and declining to interact). Children’s social cues are generally divided into *engagement* cues and *disengagement* cues. The former represents request for active interaction with a carer, and the latter represents the desire to slow down and reduce stimulation. The evaluator gave this explanation to the participants and measured the participants’ FER bias for a baby’s non-verbal communication using the BCC. The evaluator asked participants if the baby on each card indicated an engagement or a disengagement cue. We counted the number of engagement and disengagement responses and adopted the number of *disengagement* responses for the analyses. The total number of engagement and disengagement responses is invariably 52 because the total number of cards was 52. It can be thought that the larger the number of disengagement responses, the greater the sensitivity to the child’s disengagement cues. The BCC was originally developed for educational use in the parenting part of NCAST (Nursing Child Assessment Satellite Training) program [28,29,30]. BCC has shown multi-culture reliability, and the evaluators (Y.T. and K.F.) were authorized for use in the NCAST program.

#### 2.2.3. Japanese and Caucasian Facial Expressions of Emotion Task (JACFEE)

The FER performance of adults’ faces was assessed by the Japanese and Caucasian Facial Expressions of Emotion (JACFEE) [31]. The JACFEE task asks the participants to see 48 photos of adults’ faces and infer their emotional states. The participants need to identify which emotion each facial photo demonstrates from six options, that is, anger, disgust, fear, happiness, sadness, and surprise. The number of correct responses was adopted for the analyses.

#### 2.2.4. Reading the Mind in the Eyes Test (RMET)

We assessed the participants’ ability to identify the mental states of other people through photos of their eyes, using the Reading the Mind in the Eyes Test (RMET), that measures ToM [32]. The participants were asked to identify the emotional states that 36 photos of adults’ eyes exhibited. The participants were given each picture and asked to select one of four possible responses. The number of correct responses was adopted for the analyses.

#### 2.2.5. The Social Cognition Screening Questionnaire (SCSQ)

We evaluated the participants’ general social cognitive functions using the Social Cognition Screening Questionnaire (SCSQ) [33,34]. The SCSQ consists of four domains; verbal memory, schematic inference, ToM, and metacognition. The total score ranges from 0 to 40, and higher scores indicate better functional performance.

### 2.3. Measuring Methods and Psychometric Characteristics of the Tests

The EPDS and MIBQ evaluations were performed on paper, whereas the JACFEE and RMET evaluations were performed on an iPad. The BCC was performed using real cards. In the SCSQ, the evaluator posed verbal questions to the subjects.

The psychometric characteristics of the measures are as follows: for the original and Japanese version of the EPDS, the Cronbach’s α coefficients for the total score were 0.82 and 0.78, respectively [22,24]; for the original version of the MIBQ, the Cronbach’s α coefficient for the total score was 0.66 [23]; for the Japanese version of the JACFEE, the Cronbach’s α coefficient for the total score was 0.89 [35]; for the Japanese version of the SCSQ, the Cronbach’s α coefficient for the total score was 0.72 [34].

### 2.4. Statistical Analysis

Pearson’s correlational analyses were conducted to determine the associations between the scores on the EPDS and MIBQ, and scores on the social cognitive tests. We then performed two stepwise regression analyses: (1) a stepwise multiple regression analysis using the score on the EPDS as the dependent variable, and demographic data (age, gestational weeks, education level, and parity status) and scores on the social cognitive tests (BCC, JACFEE, RMET, SCSQ) as the independent variables; and (2) a stepwise multiple regression analysis using the score on the MIBQ as the dependent variable, and demographic data and scores on the social cognitive tests as the independent variables. All the statistical analyses were conducted using IBM SPSS, version 26.0.

## 3. Results

A total of 107 pregnant women were recruited for the study, however, 35 of these women refused to participate in the study. Neurologic disorders might affect social cognition, and serious obstetric co-morbidity may affect mental health. Similarly, subjects below the age of 20 years might not show adequate social cognitive development. However, none of the subjects were excluded for these reasons. Therefore, the study was conducted on the remaining 72 women who were in the second trimester. The mean age of the participants was 34.0 (SD = 4.6) years (ranging from 21 to 47), and the mean gestational weeks was 22.2 (SD = 3.5). All the participants were married and pregnant with a single child. The participants’ general and obstetric characteristics are presented in Table 1. The proportions of primipara and multipara were almost even (49% vs. 51%). The majority of the participants wanted the pregnancy, and 35% had received infertility treatment before their pregnancy. In regard to obstetric complications, 62.5% of the participants had no complications, while 32.5% had obstetric complications such as gestational diabetes. The distribution of the education level was classified as follows: 1.4% (junior high school), 12.5% (high school), 31.9% (junior college), 50.0% (university), and 4.2% (graduate school).

The mean scores on the EPDS, MIBQ, and social cognitive tests are shown in Table 2. The Pearson correlation coefficients are shown in Table 3. Positive correlations were observed between the BCC disengagement responses and EPDS scores (*r* = 0.391; *p* = 0.001), and between the BCC disengagement responses and MIBQ scores (*r* = 0.234; *p* = 0.048), suggesting that more depressed mothers and mothers with poorer bonding showed greater sensitivity to the babies’ disengagement cues. A positive correlation was observed between the JACFEE and EPDS scores (*r* = 0.237; *p* = 0.045), suggesting that more depressed mothers showed a good FER performance for adults’ facial expressions. In addition, a positive correlation between JACFEE and RMET (*r* = 0.312; *p* = 0.008) was observed.

The results of multiple regression analyses are shown in Table 4 and Table 5. In the regression analysis performed using the score on the EPDS as the dependent variable, more disengagement responses in the BCC (β = 0.365, *p* = 0.001) and primiparous women (β = −0.263, *p* = 0.016) were identified as being significantly associated with antenatal depressive symptoms (Table 4). On the other hand, in the regression analysis performed using the score on the MIBQ as the dependent variable, only disengagement responses in the BCC (β = 0.234, *p* = 0.048) were found to be significantly associated with bonding failure (Table 5).

## 4. Discussion

In this study, we attempted to evaluate the relationship between antenatal mental health and FER bias for children’s facial expressions among pregnant women, taking into consideration their broad social cognitive abilities and experience in child-raising. These results suggest that maternal sensitivity to a child’s disengagement cues might be significantly associated with antenatal depressive symptoms and bonding failure. Pregnant women who showed greater sensitivity to children’s disengagement cues were significantly more likely to have had antenatal depressive symptoms (Table 3 and Table 4), and this finding is consistent with the results of previous studies, which evaluated the association between postpartum depression and FER performance for children’s facial expressions [16,17,18]. In the meantime, the present study demonstrated an association between antenatal depression and FER bias for children’s facial expressions. This finding might imply the long-term effects of FER bias for children’s facial expressions on the longitudinal course of antenatal and postpartum depression.

The significant correlation between the scores on the EPDS and JACFEE (Table 3) supports the finding that peripartum depressive and anxiety symptoms may be associated with a greater accuracy for identifying threatening signals on adults’ faces [36]. However, the stepwise multiple regression analysis did not show an association between the scores on the EPDS and JACFEE (Table 4). Therefore, the present results indicate that antenatal depressive symptoms may show stronger association with FER bias for children’s facial expressions than with FER performance for adults’ facial expressions. Very few studies to date have assessed the FER performance for both adults’ and children’s facial expressions [17,18]. Therefore, the present study seems to have the potential strength of acknowledging the relationship between the broad context of social cognitive performance and antenatal mental health.

Moreover, our regression analysis revealed a significant association between the primipara status and antenatal depressive symptoms (Table 4). This result is also compatible with a previous Japanese study that suggested that primipara women showed a higher incidence of perinatal depression and anxiety than multipara women [37]. Our multiple regression analysis indicated FER bias for children’s facial expressions is more strongly associated with antenatal depression than with a primipara status, and this result is highly suggestive. A previous study using event-related potentials (ERP) showed that primiparous, as compared with multiparous women, showed a significantly higher neural sensitivity to infant cues [38]. Therefore, further research on the tripartite relationship between women’s mental health and neural sensitivity to infant faces and FER taking into consideration their parity, is needed.

It is noteworthy that women who showed high sensitivity to children’s disengagement cues were significantly more likely to exhibit bonding failure (Table 3 and Table 5). This novel finding could serve as a clue to unravel comorbid perinatal depression and bonding failure. This finding is nearly consistent with that of a previous study conducted using ERP, which suggested that increased early detection and processing of infant faces was associated with increased activation of the parental care system [39]. Furthermore, the relationship between abusive behavior and FER has also been clarified [19]. The results suggest bonding failure is a possible mediator of abusive behavior and FER. It is desirable to follow the longitudinal course of the effects of FER bias for children’s faces on bonding failure and abusive behavior in the postpartum period.

With regard to the ToM ability, the performance on ToM was not correlated with either the scores on the EPDS or MIBQ in the present study (Table 3). A previously conducted study in the UK examined the tripartite relationship between the FER for infant cues, ToM, and mother-to-infant bonding among non-clinical samples of mothers [20]. Our results were consistent with the results of this previous study, in which no significant relationship was observed between the performance on the ToM and score for bonding [20]. The finding of no relationship between the SCSQ and perinatal mental health also suggested that general social cognitive functions were less strongly related to maternal mental health, as compared with FER bias for children’s facial expressions.

Although the EPDS was initially developed to screen for postpartum depression, it is also widely used around the world as a screening tool for antenatal depression. In both Japan and other countries, the use of the EPDS during pregnancy has been validated for early detection of antenatal depression [40,41,42]. Moreover, a Japanese prospective cohort study revealed that the factor structure of the EPDS remains stable throughout the peripartum period [43]. In addition, a more recent Japanese study suggested that EPDS screening in the second trimester showed the highest area under the receiver operating characteristics curve to predict postpartum depression [44]. Therefore, use of the EPDS in the second trimester might be advantageous for future prospective cohort studies. Accordingly, we used the EPDS for the assessment of antenatal depression.

Some limitations of the study should be noted. Firstly, due to the small sample size and cross-sectional design, we could not conclude any causal relationships. Secondly, it is pivotal to note that the relationships between FER and mental health is complex, probably non-linear, and they may interact with other confounding variables. Thirdly, this study was conducted during the COVID-19 pandemic; the effects of the pandemic on mental status and social cognitive abilities cannot be ignored. However, our previous research indicated the scores on the EPDS and mother-to-infant bonding scale were almost consistent during the period of the pandemic in Japan [45]. Finally, since the institution that the present study was conducted in is a regional core hospital that provides advanced medical care, higher rates of infertility treatment and obstetric complications in the participants were observed. However, the other demographic data were mostly in line with those of the general population [45].

## 5. Conclusions

Maternal sensitivity to a child’s disengagement cues was associated with antenatal depression and bonding failure more strongly than other social cognitive variables, although the causal relationship still remains unclear. To clarify the impacts of FER bias for children’s facial expressions on the postpartum mental health and abusive behavior of the mother, more detailed longitudinal studies are warranted.

## Figures and Tables

**Table 1 jpm-12-01391-t001:** General and obstetric characteristics of the participants.

	Case	Percentage
**Parity**		
Primipara	35	48.6%
Multipara	37	51.4%
**Obstetric complications**		
No complications	45	62.5%
Had obstetric complications	27	37.5%
**Education Level**		
Junior high school	1	1.4%
High school	9	12.5%
Junior college	23	31.9%
University	36	50.0%
Graduate school	3	4.2%

**Table 2 jpm-12-01391-t002:** Average scores on the EPDS and MIBQ and on the social cognitive tests.

	Mean	SD
EPDS	5.5	5.0
MIBQ	3.4	2.6
JACFEE	34.4	3.8
RMET	21.4	3.1
Baby Cue Cards: Disengagement responses	20.0	7.1
SCSQ	34.9	2.5

Note: EPDS, Edinburgh Postnatal Depression Scale; MIBQ, Mother-to-Infant Bonding Questionnaire; JACFEE, Japanese and Caucasian Facial Expressions of Emotion task; RMET, Reading the Mind in the Eyes Test; SCSQ, Social Cognition Screening Questionnaire.

**Table 3 jpm-12-01391-t003:** Pearson’s correlations between the scores on the EPDS and MIBQ and on the social cognitive tests.

	(1)	(2)	(3)	(4)	(5)	(6)
(1) EPDS	-					
(2) MIBQ	0.185	-				
(3) JACFEE	0.237 *	0.090	-			
(4) RMET	−0.063	−0.020	0.312 **	-		
(5) Baby Cue Cards: Disengagement responses	0.391 **	0.234 *	0.153	−0.031	-	
(6) SCSQ	−0.044	0.014	0.105	−0.102	−0.140	-

Note: EPDS, Edinburgh Postnatal Depression Scale; MIBQ, Mother-to-Infant Bonding Questionnaire; JACFEE, Japanese and Caucasian Facial Expressions of Emotion task; RMET, Reading the Mind in the Eyes Tests; SCSQ, Social Cognition Screening Questionnaire. * *p* < 0.05; ** *p* < 0.01.

**Table 4 jpm-12-01391-t004:** Results of regression analysis conducted to examine the associations of the EPDS total score with the demographic variables and scores on the social cognitive tests.

	β	*p*	Adjusted R^2^
Baby Cue Cards: Disengagement responses	0.365	0.001 **	0.198
Parity	−0.263	0.016 *	-

Note: The EPDS total score was set as the dependent variable; the independent variables included in the stepwise regression analysis were the age, gestational weeks, parity status, education level, scores on the JACFEE, RMET, Baby Cue Cards, and SCSQ. **p* < 0.05; ***p* < 0.01; β, standardized beta; R^2^, coefficient of determination.

**Table 5 jpm-12-01391-t005:** Results of regression analysis conducted to examine the associations of the MIBQ total score with the demographic variables and scores on the social cognitive tests.

	β	*p*	Adjusted R^2^
Baby Cue Cards: Disengagement responses	0.234	0.048 *	0.041

Note: The MIBQ total score was set as the dependent variable; the independent variables included in the stepwise regression analysis were the age, gestational weeks, parity status, education level, scores on the JACFEE, RMET, Baby Cue Cards, and SCSQ. * *p* < 0.05; β, standardized beta; R^2^, coefficient of determination.

## Data Availability

The data presented in this study will be made available by the corresponding author upon reasonable request. The data are not publicly available due to privacy restrictions.

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
