# Peer review of "Relationship between Antenatal Mental Health and Facial Emotion Recognition Bias for Children’s Faces among Pregnant Women"

_jpm, 2022, doi:10.3390/jpm12091391_

Round 1

Reviewer 1 Report

This was a clearly articulated manuscript on an interesting and worthwhile subject. I have only a couple of comments.

1) It would be useful to add the reasoning behind including women in the second trimester

2) If parity is so important, would it be worthwhile performing the correlations on each group (primi, multi) separately?

Reviewer 2 Report

The aim of this cross-sectional study was to determine the relationship of facial emotion recognition bias for children’s faces with antenatal depression and bonding failure among pregnant women. For reaching this purpose, authors have conducted study on 72 pregnant women in their second trimester via using depression and social cognition scales. They concluded that maternal sensitivity to the child’s disengagement cues was associated with antenatal depressive symptoms and bonding failure more than the other social cognitive variables. The study is a valuable study focused on an interesting topic. It needs some revisions and language editing;

1. The abstract should state the purpose of the study. 

2. The hypothesis of the study and the gap it fills in the literature should be included in the introduction. 

3. The inclusion and exclusion criteria of the study are not clear. In the exclusion criteria, it should be explained which neurological diseases are exclusion criteria and why this criterion was set. Likewise, what is meant by serious comorbidity should be given. The rationale for excluding people under the age of 20 should be stated. How many people were excluded for these reasons should be added. 

4. It should be clear which tests were performed on ipad and which tests were performed on paper.

5. Psychometric properties and Cronbach's alpha values of both the original and Japanese versions of the administered tests should be included.

6. Why was the Edinburgh Postnatal Depression Scale preferred? How sensitive is this scale to antenatal depressive symptoms? Does the sensitivity reported in other cultures apply to the Japanese population? This should be addressed in the discussion.

7. The first sentence in the discussion was quite assertive and should be softened.

8. The discussion is like a repetition of the findings, more comparison with the literature, interpretation, and analysis should be added. . 
